

# Schrödinger's yeast: the challenge of using transformation to compare fitness among *Saccharomyces cerevisiae* that differ in ploidy or zygosity

Linnea Sandell[1], Stephan G. König[1,2] and Sarah P. Otto[1]

[1] Department of Zoology and Biodiversity Research Center, University of British Columbia, Vancouver, Canada
[2] Department of Computer Science, University of British Columbia, Vancouver, British Columbia, Canada

## ABSTRACT

How the number of genome copies modifies the effect of random mutations remains poorly known. In yeast, researchers have investigated these effects for knock-out or other large-effect mutations, but have not accounted for differences at the mating-type locus. We set out to compare fitness differences among strains that differ in ploidy and/or zygosity using a panel of spontaneously arising mutations acquired in haploid yeast from a previous study. To ensure no genetic differences, even at the mating-type locus, we embarked on a series of transformations, which first sterilized and then temporarily introduced plasmid-borne mating types. Despite these attempts to equalize the haplotypes, fitness variation introduced during transformation swamped the differences among the original mutation-accumulation lines. While colony size looked normal, we observed a bi-modality in the maximum growth rate of our transformed yeast and determined that many of the slow growing lines were respiratory deficient ("petite"). Not previously reported, we found that yeast that were *TID1/RDH54* knockouts were less likely to become petite. Even for lines with the same petite status, however, we found no correlation in fitness between the two replicate transformations performed. These results pose a challenge for any study using transformation to measure the fitness effect of genetic differences among strains. By attempting to hold haplotypes constant, we introduced more mutations that overwhelmed our ability to measure fitness differences between the genetic states. In this study, we transformed over one hundred different lines of yeast, using two independent transformations, and found that this common laboratory procedure can cause large changes to the microbe studied. Our study provides a cautionary tale of the need to use multiple transformants in fitness assays.

## INTRODUCTION

From the very first description of mutations in the form of newly arisen polyploids in plants by *de Vries (1909)* there have been disputes regarding the phenotypic and fitness

Corresponding author
Linnea Sandell,
nhl.sandell@gmail.com

effects of mutations (*Eyre-Walker & Keightley, 2007*). Importantly, mutations seem to have different effects depending on the genetic background in which they appear (*Chandler et al., 2014*)—for sexual species with a strict alternation of ploidy, selection may act on a mutation to different degrees or potentially in opposite directions in the haploid or diploid phase. We set out to test the fitness of mutant haplotypes in different genetic states.

It is standard practice in molecular biology to construct vectors carrying genes of interest and transforming these into bacteria or yeast (*Gietz & Woods, 2001*; *Gietz & Schiestl, 2007*). For transforming the plasmid into cells, we use a monoculture (a culture arising from a single cell), and as such, we work under the assumption that all colonies growing on our selective plates after transformation are of the same kind. Any one colony can hence be used to analyse the phenotypic effect of carrying and expressing the plasmid. In this study we transformed over one hundred different lines of yeast, and found that this common laboratory procedure can cause large changes to the microbe studied. Our study is a cautionary tale of the need to use more than one transformant in phenotypic assays. Portions of this text were previously published as part of a thesis (https://open.library.ubc.ca/media/stream/pdf/24/1.0397308/3).

## Motivation for study: ploidy specific effects of mutations

Among the many distinguishing characteristics of life is the number of copies of an organism's genome ("ploidy"). While many organisms spend the majority of their life with one copy of their genome (most bacteria but also multicellular haplont organisms such as dictyostelid slime moulds and green alga of the genus *Chara*), others have two (most animals), or many (polyploid plants). Yet others exist as multicellular forms in both ploidy stages, called haploid-diploid or haplodiplontic (many green, brown, and red macroalgae). Even in multicellular diplont organisms such as humans, the difference in selection in the haploid and diploid cell are of great importance. For example, mutations that render fitness advantages in gametes may have unpredicted consequences for a developing zygote (*Immler, 2019*).

Studies measuring differences in the reproductive fitness of haploid and diploid stages of the life cycle have discovered a strong interactive effect of the environment (*Gerstein & Otto, 2009*; *Mable, 2001*; *Mable & Otto, 1998*; *Thornber, 2006*; *Zörgö et al., 2013*). Many of these studies are conducted in budding yeast, *Saccharomyces cerevisiae*, that can be maintained in both its haploid and diploid form. The researchers have reported broad differences in growth rate or morphology of the two ploidy levels. Nevertheless, few studies have been conducted to measure mutational effects on fitness in the two different states.

*Gerstein (2012)* compared the fitness and phenotypic effect of 40 different nystatin-tolerant mutations in *S. cerevisiae* and found that haploids acquired stronger resistance to the toxin and acquired greater increases in growth rate compared to their homozygous diploid counterparts. *Szafraniec et al. (2003)* measured differences in growth rate of mutant haploids created by mutagenesis and heterozygous diploids but did not report a comparison of the homozygous diploids to haploids. Here, we ask if fitness effects of spontaneous mutations from a mutation-accumulation (MA) experiment (*Sharp et al., 2018*) differ between ploidy levels.

## Dominance

The genetic dominance of a mutation describes the difference in phenotype with or without the presence of a functional version. There are mutations that are imperceptible in the heterozygous state but fully expressed in the homozygous state and as such have a dominance of 0. The dominance coefficient, $h$, determines the likelihood that a beneficial allele spreads and fixes in a population (known as Haldane's sieve, *Haldane (1927)*). Conversely, the frequency of deleterious mutations at mutation-selection balance will be inversely proportional to the dominance coefficient (*Haldane, 1937*). Whether through population structure (*Whitlock, 2002*) or inbreeding (*Morton, 1971*), the dominance coefficient will determine the mutation load in populations with non-random mating. Understanding the average as well as the variance in dominance of spontaneous mutations is of interest in both conservation biology and medicine.

There is also reason to believe that dominance effects may correlate with mutational effect and type of protein. The shape of this relationship determines the possible mutational load and inbreeding depression of a species. While outside the scope of this study, the underlying mechanism for differences in dominance is one of the oldest and most debated topics in evolutionary biology (*Billiard & Castric, 2011*).

In addition to determining fixation probabilities, mutation frequencies, and the impacts of inbreeding, the dominance coefficient can explain the prevalence of haploid-diploid life cycles. We should expect a diploid-dominant life cycle if deleterious mutations are sufficiently recessive in diploids (*Perrot, Richerd & Valéro, 1991*), while a haploid-dominant life cycle is expected when deleterious mutations are dominant (*Jenkins & Kirpatrick, 1994*, *1995*). While it is commonly assumed that the mutational effect on fitness in a haploid is equal to that of a diploid homozygous for the mutation, recent theoretical work has demonstrated that intrinsic differences in how mutations affect the haploid and diploid stage of a life cycle can explain the stable co-existence of haploid-diploid organisms (*Scott & Rescan, 2017*). By assigning separate fitness effects of mutations in the homozygous diploid ($s_d$) and haploid ($s_h$) phase, the authors identified conditions under which predominantly haploid ("haplont"), predominantly diploid ("diplont"), or biphasic life cycles ("haploid-diploid") might evolve.

There have been several attempts to measure the distribution of dominance coefficients (Table 1 in *Agrawal & Whitlock, 2011*; and Table 1 in *Manna et al., 2012*). Data from the Yeast Knockout Collection (YKO) (*Shoemaker et al., 1996*) were used to estimate the distribution of dominance coefficients (*Phadnis & Fry, 2005*; *Agrawal & Whitlock, 2011*; *Manna et al., 2012*), revealing a negative correlation between selection strength and dominance. One limit of knockout data is that they are, by definition, large effect mutations with a complete loss of function in the homozygous diploid and do not represent the multitudes of small effect mutations that are likely to arise spontaneously. As *Manna et al. (2012)* pointedly note, the YKO was not created to answer questions of dominance. In addition, these studies have not included organisms of different ploidy levels. There is a need for datasets of mutational effects measured with high replication and precision in haploids, heterozygous, and homozygous diploids to help parameterize models of life cycle evolution.

## Our contribution

In this study, we use previously described MA lines (*Sharp et al., 2018*) to produce lines of different ploidy and copies of the MA genome: from haploid lines to heterozygous and homozygous diploid. Our goal was to measure how selective effects in haploids correlate with the selective effects in homozygous diploids, while at the same time measuring the average of and variance among dominance effects in heterozygous yeast.

## Justification for method

A commonly used method to produce homozygous diploids from haploids is by mating type switching. Haploid yeast have two different mating types, *MAT*a and *MAT*α. All yeast have both mating type loci present in their genome, but one is inactive. Wild yeast are "homothallic" and can, *via* gene conversion using an endonuclease called HO, switch which mating type is active. Homothallism makes it possible for haploid cells in the absence of mating partners to switch the mating type they express and mate with a neighboring cell. In contrast, laboratory strains of yeast are typically heterothallic and cannot switch mating type, due to a deletion mutation of the *HO* locus. To produce homozygous yeast, researchers can insert the *HO* locus on a plasmid into the haploid yeast, which will readily switch mating type and mate. The problem with using this approach to compare mutational effects across ploidy levels is that the diploid that is formed will have a different mating type, namely *MAT*a/*MAT*α, and will therefor differ in more than genomic copy number (*Birdsell & Wills, 1996*).

An alternative approach, used by *Gerstein (2012)*, is to temporarily transform *MAT*a haploids with a plasmid containing the opposite mating type. With sufficient copies of the *MAT*α plasmid the transformed haploid yeast will act like a *MAT*α haploid and mate readily with untransformed *MAT*a cells. Through propagation post mating, the plasmid with the added mating type is lost, and the researcher has acquired a diploid homozygote of the *MAT*a/*MAT*a genotype. This method is not without risk, however, as it can allow for further mating of the *MAT*a/*MAT*a diploids to haploids still possessing the *MAT*α-containing plasmid, leading to lines with higher ploidy levels.

In this experiment, we introduce a new method. By deleting both the *MAT* locus and the *STE4* locus we rendered haploid yeast sterile. We then temporarily transform the yeast with two different kinds of plasmids that contain the lost genes and restore mating ability as *MAT*a or *MAT*α. Once mating has occurred, the diploid is selected to lose the plasmid, leading to a *mat*Δ0/*mat*Δ0 mating type.

Unfortunately given our goals, in the present study transformations induced large changes in fitness that obscure the smaller mutational effects we wanted to measure. In particular, we found high rates of loss in respiratory function, also known as petiteness (respiratory deficiency known to decrease growth rate) in yeast. Even when considering sets of yeast lines with the same petite status, we found aberrant growth patterns across the haploid, heterozygous, and homozygous states. This indicates that our experimental lines changed in more ways (genetically, epigenetically, or otherwise) than those easily scored by petiteness and that these changes have greater impacts on growth rate than the mutations in the MA lines. We conclude that the haploid, heterozygous, and homozygous genotypes

of the MA lines have acquired new variation, preventing us from extracting any information on the dominance of the small-effect spontaneous mutations acquired during MA. This conclusion highlights the challenge of estimating the effects of spontaneous mutations across ploidy levels: methods designed to hold "all else equal" instead induce differences.

## METHODS

The laboratory notebooks, raw data, and statistical analyses are available on the GitHub site associated with this manuscript (https://github.com/LinneaSandell/schrodingers-yeast.git).

### Strain creation

We used 100 haploid lines of both mating types that had gone through mutation accumulation (*Sharp et al., 2018*), as well as 33 replicates of their SEY6211 ancestor as controls (*MATa/α, ho, leu2-3 112, ura3-52, his3-Δ200, trp1-Δ901, ade2-101, suc2-Δ9*). Half of the lines were *rdh54Δ::KANMX*. The *RDH54* gene is involved in recombinational repair (*Klein, 1997*), and its effect on mutation accumulation in diploids was studied in the original MA study (*Sharp et al., 2018*). The lines were streaked out from frozen on yeast peptone glycerol (YPG) agar plates to verify that the lines had functional respiratory pathways. Single colonies were then streaked out in patches on yeast peptone dextrose plates supplemented with additional adenine (YPAD) to inhibit reversion of the *ade2* mutation (*Achilli et al., 2004*).

### Transformation protocol

We followed the LiAc/SS carrier DNA/PEG method described by *Gietz & Schiestl (2007)*. We grew overnight cultures from single colonies that were used to grow competent cells (cultures in exponential phase of growth). The culture of competent cells was washed with sterile water and a lithium acetate solution before we added transformation mix (containing the PCR product of interest, ssDNA, 50% PEG, water and 1 M lithium acetate). The cells were heat shocked with the transformation mix for 1 h at 42 °C. The cells were then spun down and resuspended in water before being plated on selective media corresponding to the transformation at hand.

#### Step 1: deletion of the STE4 locus

To avoid mating during transformation of our *MATα* lines into Δ*mat* transformants (that act as *MATa* and therefore could mate with their non-transformant *MATα* siblings (*Strathern, Hicks & Herskowitz, 1981*)) we chose to make our lines sterile by deleting the *STE4* locus and replacing it with the *TRP1* gene (for which our original lines were deletion mutants). We amplified the *TRP1* locus from the pFA6a-*TRP1* plasmid (a gift from John Pringle: Addgene plasmid #41595; http://n2t.net/addgene:41595; RRID:Addgene_41595, *Longtine et al., 1998*), using primers constructed to contain homologous sequences upstream and downstream of *STE4* and bind to regions flanking *TRP1* on the plasmid (see Table S1 for primer sequences). The resulting PCR product was used to replace *STE4* with *TRP1*, see Fig. 1A.

## A. Knockout of MAT and STE4 loci

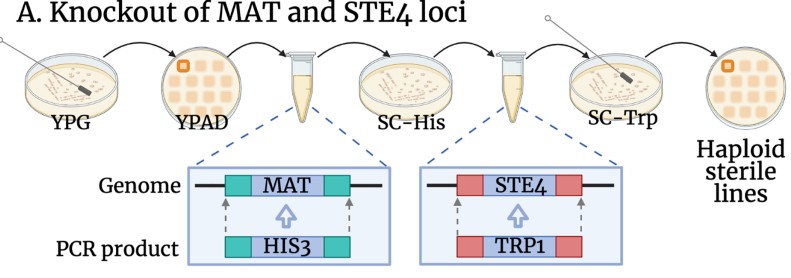

## B. Plasmid transformation with OLP003 and OLP004

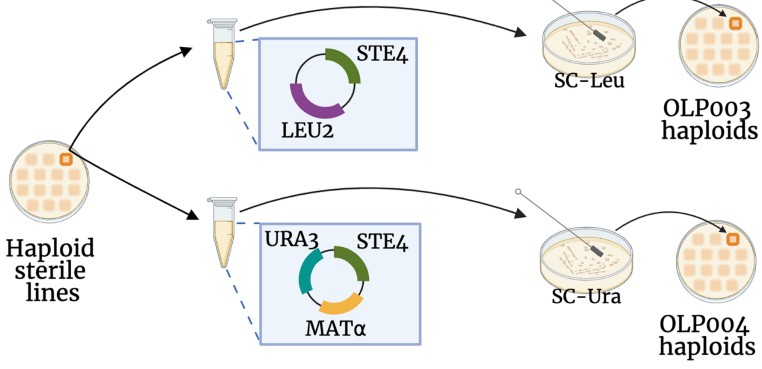

## C. Mating and propagation for loss of plasmids

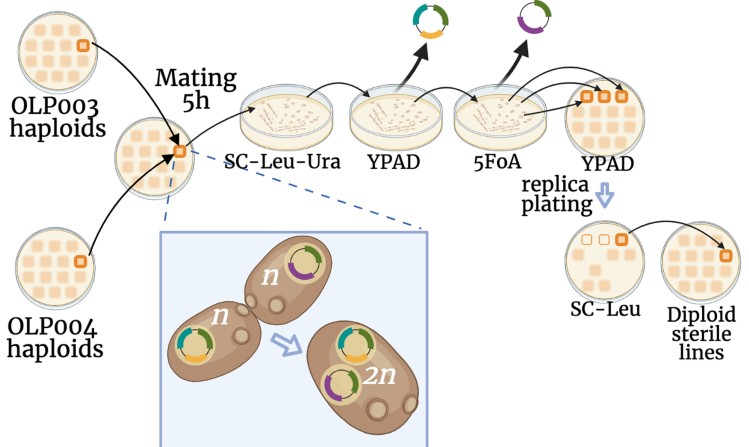

**Figure 1 Transformations conducted in this experiment.** The fitness of diploid sterile lines (C) were compared to haploid sterile lines (A) in the first transformation experiment (DS1) and to OLP003 or OLP004 haploids (B) in the second (DS2). Created with BioRender.com.

Full-size ⬚ DOI: 10.1371/journal.pgen.1003388

Transformation was conducted using 1-h heat shock at 42 °C. Yeast that were successfully transformed and had their *STE4* locus replaced by *TRP1* were selected through growth on agar plates with synthetic complete medium lacking tryptophan (SC-Trp). A single colony from the SC-Trp plate was then streaked to single colonies again on SC-Trp. These colonies were used to verify the knockout through PCR (see laboratory notebooks on GitHub for gel pictures) and to proceed with strain creation.

### Step 2: deletion of the MAT locus

Using similar transformation, we deleted the *MAT* locus in our lines and replaced with the *HIS3* gene (for which our original lines were deletion mutants). We amplified the *HIS3* locus from the pFA6a-*His3MX6* plasmid (a gift from John Pringle: Addgene plasmid # 41596; http://n2t.net/addgene:41596; RRID:Addgene_41596, *Longtine et al., 1998*). The primers used were constructed to contain homologous sequences upstream and downstream of *MAT* and bind to regions flanking *HIS3* on the plasmid (see Table S1 for primer sequences). The resulting PCR product was used to delete the *MAT* locus with *HIS3*, see Fig. 1A.

Transformation was conducted using 1-h heat shock at 42 °C. Yeast that were successfully transformed and had their *MAT* locus replaced by *HIS3* were selected through growth on agar plates with synthetic complete medium lacking histidine (SC-His). A single colony from the SC-His plate was then streaked to single colonies again on SC-His. These colonies were used to verify the knockout through PCR (see laboratory notebooks on GitHub for gel pictures). Once confirmed, a single colony from the plate was inoculated into 2 mL of YPAD media, grown at 200 rpm at 30 °C to saturation, and frozen in 15% glycerol (500 µL of saturated culture mixed with 500 µL of 30% glycerol). One line (line ID 52, line 98 in *Sharp et al. (2018)*) that had accumulated a mutation in *HIS4* in the MA experiment (an A to T mutation in pos chrIII:67039 leading to a premature stop codon) failed to grow on the SC-His plate post transformation and was dropped from the experiment. The 132 lines that had both the *MAT* and *STE4* locus deleted were given new identifiers as knockout (KO) lines, grown to saturation and frozen in 15% glycerol. These lines were used in Step 3 below, as well as in dataset 1 (DS1).

### Step 3: MA to KO comparison

To control for off-target insertion during the knockout of the *MAT* and *STE4* loci we ran growth rate assays for the lines before and after the knockout procedure. We grew both the original lines with their ancestors and the lines with the deletions. A total of 20 µL of frozen stock of these lines were inoculated in 2 mL YPAD media at 30 °C at 200 rpm for 2 days, after which the saturated culture was transferred to microcentrifuge tubes and stored at 4 °C, where they were kept for the duration of the growth rate assay. The lines were randomized across days and wells in the growth assay. To set up the growth assay, each culture was diluted 1:121 with YPAD medium. The cultures were grown for up to 24 h, at 30 °C with continuous shaking set on Medium on a Bioscreen C machine. The optical density (OD) at 600 nm wavelength of the cultures was measured every 15 min. We conducted 11 replicate measurements of each line (the raw data files as well as computed growth parameters are available on GitHub).

We used the spline fitting method to extract the maximum slope of the growth curve, as described by *Gerstein (2012)*. We ignore OD measurements before 45 min to allow the culture to reach the set temperature and be thoroughly mixed. In our models, initial OD was set to the mean OD from 45 min to 2 h (5 OD measurements). Initial OD is taken as a proxy for the quality of the medium (a higher OD of the blank medium corresponds to more caramelization during autoclaving, leading to less accessible nutrients).

We computed the difference in mean growth rate of each line before and after knockout of *STE4* and *MAT* and flagged four that had a z-score below −2.5. The analysis presented exclude these lines.

To estimate the average effect of the *STE4* and *MAT* knockout we fit a linear mixed effect model of maximum slope as a function of KO (wildtype or knockout of *STE4* and *MAT*), MA (MA or control), *RDH54* status (wildtype or deletion mutant), and initial OD. We included day and machine as random effects. Each line had a line identity number (line ID) from the previous mutation-accumulation experiment, which is used as a random effect, and is the same for the line before and after knockout. The presence of genetic variance in the data is verified by fitting a simpler model (excluding KO and MA) to each of four groups: the control lines and MA lines before and after knockout. The significance of the genetic variance in explaining the data is evaluated by analysis of variance (ANOVA) of the model including line ID and a model excluding it.

### Step 4: plasmid transformation of the haploid lines

All KO lines were separately transformed with two plasmids: OLP003 contains the *STE4* and *LEU2* loci, OLP004 contains the *MATα*, *STE4*, and *URA3* loci (see Fig. 1B). A single colony from each plasmid-transformed line was streaked to grow on a YPAD plate, from which we sampled haploid lines used to create the heterozygous and homozygous diploids (see below).

The derivation of diploid lines was conducted twice, which we refer to as dataset 1 (DS1) and dataset 2 (DS2). In DS1, growth in diploids was compared to that of the haploid line after knockout. In DS2, each line was randomly chosen (with two exceptions, see below) to have undergone OLP003 transformation or OLP004 transformation to represent their haploid state. We sampled the chosen plate with a toothpick and streaked it out to single colony on YPAD twice, after which three colonies were patched on both YPAD and leucine (in the case of OLP003 transformant) or uracil (for OLP004 transformant) drop-out plates. The colonies that grew on YPAD but not leucine/uracil drop-out plates were understood to have lost the plasmid and inoculated in YPAD, grown for 2 days in a 30 °C shaking incubator and then frozen into aliquots with 15% glycerol for use as haploid comparison lines.

One line (line ID 39 this study, line 81 in *Sharp et al. (2018)*) failed to be transformed with OLP004. We chose the OLP003 transformant to represent this line's haploid state. One line (line ID 83 this study, corresponding to line 157 in *Sharp et al. (2018)*) with a preexisting mutation in *LEU3* (a G to T mutation at position chrXII:1036202, mutating the cysteine to a phenylalanine) failed to grow on the SC-Leu plate post transformation with OLP003. We suspect *LEU2* inserted during the *MAT* locus deletion was insufficient to rescue this mutant. We chose the OLP004 transformant to represent this line's haploid state. One line (line ID 84 this study, corresponding to line 158 in *Sharp et al. (2018)*) could not be successfully transformed with either plasmid and was dropped from the experiment. The point mutations for all lines are available in the supplemental material of *Sharp et al. (2018)*.

### Step 5: mating

To create the heterozygote state, a swab from the OLP003 transformant was patched on a plate together with a random OLP004 transformant from one of the control lines. For the homozygote state, the swab from the OLP003 transformant was patched on a plate together with its OLP004 transformant. The lines were allowed to mate for 5 h before being transferred and spread out with a cotton tip on double drop-out SC-Ura-Leu plates, see Fig. 1C. After 2 days of growth a colony growing on the double drop-out was picked and streaked out on YPAD. After another 2 days of growth a single colony was chosen and streaked out on an agar plate containing 5-Fluororotic acid (5FoA), on which only yeast with a deficient uracil pathway can grow, to confirm loss of the OLP004 plasmid. After 3 days of growth, three colonies were taken and patched on YPAD and SC-Leu, to confirm loss of the OLP003 plasmid. After 2 days, a colony that grew on YPAD but not on SC-Leu was inoculated in YPAD, grown for 2 days in a 30 °C shaking incubator and then frozen into aliquots with 15% glycerol.

### Step 6: fitness and ploidy assays

The growth rates of DS1 were measured together with the double knockout haploid lines prior to their transformation in August 2019. The growth rates of DS2 were measured together with haploids after their plasmid transformation in January 2020. We conducted the growth rate assays as described in Step 3: MA to KO comparison.

We confirmed the ploidy of our lines by flow cytometry. We used a protocol modified from *Nash et al. (1988)* described by *Gerstein et al. (2006)*. Samples were inoculated from fridge cultures (15 μL into 1 mL of YPAD with added 50 μg mL$^{-1}$ of ampicillin in 96-well boxes). After one night of growth 5 μL of each culture was transferred to 96 well assay plates and washed in water before being fixed in 70% ethanol. The samples in ethanol were either kept at room temperature for over 1 h or overnight in the fridge before proceeding. Plates were then centrifuged at 2,500 RPM for 5 min, washed in 50 mM sodium citrate solution, and then incubated at 37 °C in a 50 mM sodium citrate solution with 6.25 μg of RNAase A added to each sample. The following day the samples were spun down, supernatant removed, and the cells stained in a solution of sodium citrate and 7.5μL of 50 mM sytox green per sample. The stained samples were run on an Attune NxT flow cytometer, which allows processing of 96-well plates. We ran the samples at 25 μL min$^{-1}$ until 10,000 events had been measured. The resulting files were exported and analyzed with the flowCore package in R (the raw data files as well as summary figures are available on GitHub). All lines had the expected ploidy level.

## RESULTS

### Knockout of the *MAT* and *STE4* knockout lead to increased fitness, and new variation in growth rates

Four MA lines fell 2.5 standard deviations below the average knockout effect, with large decreases in their growth rates post knockout (line IDs 2, 106, 122 and 127). We exclude these lines in the comparison of lines before and after knockout. The knockout of *MAT*

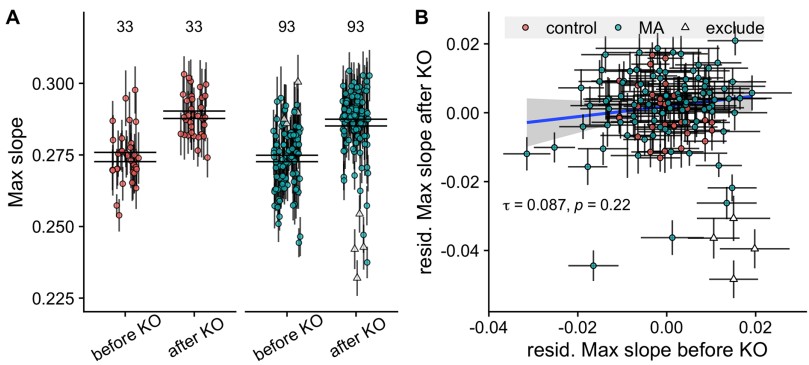

**Figure 2 Comparison of growth rates before and after knockout of the *STE4* and *MAT* loci.** (A) Maximum growth rates of control and MA lines increase after knockout of the *STE4* and *MAT* loci. Large black lines represent the standard error of the mean in each group. Numbers above plot show number of lines in each group. (B) Correlation of growth rates (max slope) of MA lines before and after knockout. Triangles signify the four MA lines whose change in growth rate fell more than −2.5 standard deviations below the mean and were excluded from the analysis. Colored points with grey error bars represent line means and standard error of the mean (calculated from 11 replicate fitness assays).

and *STE4* led to increases in growth rates of both control and MA lines (t-value 15.87, $\chi^2$ of model with compared to without effect of KO = 241.4, $P < 10^{-5}$), see Fig. 2A.

We found a significant difference in growth rate among MA lines, but not control lines, before knockout ($\chi^2 = 42.3$, $P < 10^{-5}$ for MA lines, $\chi^2 = 3.2$, $P = 0.073$ for control lines). After knockout of the *STE4* and *MAT* loci, we verified this result ($\chi^2 = 195$, $P < 10^{-5}$ for MA lines, $\chi^2 = 1.94$, $P = 0.16$ for control lines).

Despite the promising variance in growth rates among MA lines we found no correlation between the growth rate of lines before and after knockout (Kendall's rank correlation $\tau = 0.087$, $P = 0.22$, see Fig. 2B). This is problematic because it indicates that whatever signal existed before about the impact of mutations on growth had been swamped by variation induced by the knock-outs.

## High frequency of respiratory incompetent yeast ("petites") following plasmid transformation

In an initial experiment (DS1), we generated diploids by combining two plasmid-transformed haploids, followed immediately by selection for plasmid loss. We found a large reduction in the average growth rate of these diploid lines, compared to the corresponding haploid lines, but the latter had not undergone transformation with OLP003 or OLP004. To determine if this plasmid transformation was the source of the large decreased fitness of the diploids, we re-transformed our lines, saving the haploid lines after transformation to compare the growth of these to the diploids created from them (DS2). On average, we found that the haploids had also decreased in growth rate compared to DS1. However, we also found very strong bimodality in our data, see Fig. 3.

To determine whether variation among lines might have been caused by loss of mitochondrial function we spotted our lines on YPG to test for respiratory competence. While the number of petites was low among haploids in DS1 (seven out of the 131 lines

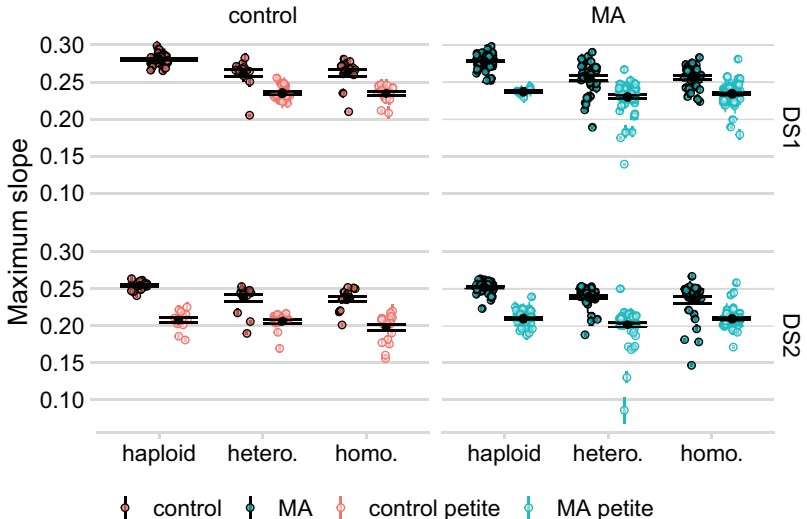

**Figure 3 Respiratory ability of the line strongly predicts maximum slope of growth curve in both datasets.** Large black points and lines represent the standard error of the mean in each group. Colored points with error bars represent line means and standard error of the mean (calculated from 11 replicate fitness assays).

**Table 1 Kendall's correlation coefficient of mean growth rate of MA lines with same petite status in DS1 and DS2.**

| Genotype | Petite status | Correlation coef | P-value | # |
|---|---|---|---|---|
| Haploid | Grande | 0.184 | 0.077 | 64 |
| Heterozygote | Petite | −0.108 | 0.309 | 57 |
| | Grande | −0.076 | 0.679 | 27 |
| Homozygote | Petite | 0.092 | 0.342 | 63 |
| | Grande | 0.033 | 0.894 | 23 |

without plasmid transformation: four of these were the four lines we detected as outliers in our comparison of growth rates before and after knockout), we found a high prevalence in the diploids (169 out of 260 lines). In DS2, where haploids were isolated post plasmid transformation, we found about an equal proportion of petites in the haploid and diploid lines (about half of our lines were petite in both groups). After further analysis, we found that diploid lines that had the *RDH54* gene intact were more likely to become petite ($\chi^2$ for DS1 = 36, $P < 10^{-4}$, $\chi^2$ for DS2 = 61, $P < 10^{-4}$). We also found that the frequency of petites was significantly higher in the haploids that were transformed with OLP004 (carrying *STE4* and *MATα* and *URA3*) compared to the haploids that were transformed with OLP003 (carrying *STE4* and *LEU2*) ($\chi^2 = 65$, df = 1, $P < 10^{-5}$).

## No correlation in growth rates across data sets

We found no significant correlation between the growth rate of lines in DS1 and DS2 with the same genotype and petite status (Table 1). Hence, we analyzed the two datasets separately.

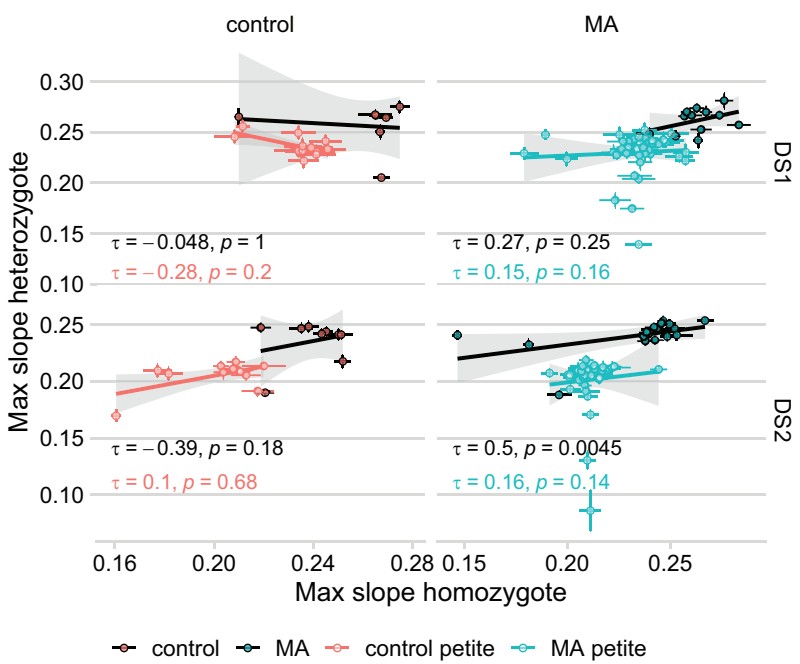

**Figure 4 Growth rate of the MA line in its heterozygous and homozygous form are not significantly correlated.** Thick lines represent Kendall's rank correlation for each group. Colored points with error bars represent line means and standard error of the mean (calculated from 11 replicate fitness assays). The significant correlation in MA lines of DS2 remains even when removing the seeming outlier.

## Inferences of mutational effects across haploid, heterozygote, and homozygote yeast

The linear mixed effect models we fit to DS1 and DS2 have the same structure: maximum slope is fitted as a function of petite status, *RDH54* status, MA status (whether MA or control), genotype, and initial OD as fixed effects. We also include an interaction between MA status and genotype. We control for day, machine (in DS1 only, as all DS2 replicates were measured on one machine), and haplotype effect (an identifier shared by the haploid, heterozygote, and homozygote from the same haploid genome) by including them as random effects in our model. Neither MA nor the interaction of MA and genotype is significant in DS1 (t-value = −1.45, $P$ = 0.15 for MA). Haploid lines had a significantly higher growth rate than diploids in DS1 (t-value = −14.163 and −13.950 for heterozygotes and homozygotes respectively, $P = 10^{-5}$ for both). In DS2 MA is also not significant, but the interaction between MA and homozygosity is (t-value = 5.78, $P = 10^{-5}$), with homozygous MA lines having a higher average growth rate than homozygous control lines.

Because we found no correlation between how diploid lines of the same petite status grew in DS1 and DS2 (Table 1), we considered the possibility that unique fitness-affecting mutations occurred in each line during transformation. In effect, this would mean that the aim of our study, to compare the growth rate of identical mutated genomes in three different genotypic conditions, would be compromised. If this were the case, we would not expect a correlation between the fitnesses of the haploid and diploid homozygous state of
each haplotype, because the fitness effects of pre-existing mutations would be swamped by new and different mutations in the two states. We compared our original model that used MA identity as a random effect on intercept (with the same identity for the haploid and homozygous lines derived from the same MA line) to a model using line identity as a random effect (that treats each genotype as its own group). In this latter model, there is no information on the original haplotype used to establish the line, but the fixed effects remain the same (petite status, *RDH54* status, MA status (whether MA or control), genotype, and initial OD). The model using line identity rather than haplotype is preferred in both DS1 ($\Delta$AIC = −1,021) and DS2 ($\Delta$AIC = −2,579), suggesting that our MA mutations are overwhelmed by unique fitness-affecting mutations that occurred in each line during transformation. These results remained unchanged when excluding the four lines that were outliers in the KO to MA comparison. Our findings are evident also in plotting the correlation of heterozygote and homozygote fitness (Fig. 4): we find a significant correlation only in the grande MA lines in DS2. The fact that the correlation was not found in DS1 further emphasizes the lack of repeatability in our experiment.

## DISCUSSION

The fitness effects of spontaneous mutations depend on the genotypic state of the organism. While it is often recognized that a deleterious mutation will not be fully recessive (*i.e.*, heterozygote mutants will differ from wildtype), less attention has been paid to differences in the fitness effect of mutations across ploidy levels (*i.e.*, haploid mutants may differ in fitness from homozygote mutant diploids). Previous studies have been limited to single genes or considered only one aspect of the question (comparing only heterozygous and homozygous mutants, or only haploid and diploid mutants). In this project, we set out to conduct a large-scale experiment to measure both dominance within diploids and differences between ploidy levels using 100 previously established haplotypes with fitness-affecting spontaneous mutations in yeast (*Sharp et al., 2018*). By establishing haploid, heterozygote, and homozygote genotypes of each haplotype, we aimed to study the fitness effect of mutations in each of these genotypic states with enough replicated fitness assays that the fitness of each line could be accurately measured. In addition, we disrupted the mating pathway in all of our yeast, to control for differential expression between mating types.

To establish the three genotypic states of our mutant genomes (bearing MA haplotypes) we put our lines through a number of transformations. These transformations led to high rates of loss of respiratory function (petites), accompanied with low growth rates. The combination of heat shock (*Van Uden, 1985*) and serial bottlenecks (*Taylor, Zeyl & Cooke, 2002*) potentially account for the high frequency of petites. There is also evidence that the *his3-Δ200* allele carried by our MA ancestor increases the formation of petites under heat stress (*Zhang et al., 2003*). The loss of respiratory function is an easily distinguished mutant phenotype yielding smaller colonies on non-fermentable carbon sources, and the genetic underpinnings can be numerous (*Dimitrov et al., 2009*). Because we grew our lines on fermentable media, they were not originally caught. Even if the petite mutation was the same across our experimental replicates, the fitness interaction between

mitochondrial and nucleic mutations is often significant (*Zeyl, Andreson & Weninck, 2005*). Four independent studies have assayed the YKO of 4,800 non-essential genes (*Giaever et al., 2002*) for respiratory competence (*Dimmer et al., 2002*; *Merz & Westermann, 2009*; *Luban et al., 2005*; *Stenger et al., 2020*). Remarkably, these studies reveal a surprisingly low degree of overlap. Out of the 563 genes reported to exhibit the petite phenotype, only 113 were consistently identified in all four studies (*Stenger et al., 2020*). This divergence in results may, in part, be attributed to the inherent variability of the petite phenotype, even within the same genetic background.

Our finding of non-intentional effects of transformations mimic those found in the YKO (*Giaever et al., 2002*). They inferred that 26% of transformed diploids carried an off-target mutation, and 6% of their transformed strains had a noticeable reduction in growth that was not explained by the gene deletion itself. Following the knockout of *MAT* and *STE4* we found four lines with severely reduced growth (4/131 = 3.05% so around 1.5% per transformation). Notably, the 6% observed in the YKO was from knockouts of 5,916 different genes while we did 123 independent knockouts of the same two genes. Other studies have argued that it might not be the transformation itself that is mutagenic, but rather, the effect of a gene knockout that favours compensatory mutations to arise (*Teng et al., 2013*). Indeed, *Friedman et al. (2018)* reports low off-target mutations in *Cryptococcus neoformans*, while *Teng et al. (2013)* find a large amount of genetic heterogeneity within the YKO lines.

Even if we cannot know for sure to what extent the decreased growth we observed was due to non-intended mutations arising during the transformations, and compensatory mutations following the knockout of *STE4* and *MAT*, we tend to favour the first alternative. This is primarily because of the set of haploids that went through the knockout of *MAT* and *STE4*, but not through plasmid transformation. This set of haploids were included in two separate growth assays: once to determine the effect of knockout and once in DS1. These assays were run two months apart with independent revivals from freezer stocks. *Teng et al. (2013)* reports the genetic heterogeneity from secondary mutations arising already in a single plating from the freezer, showing big inter-colony variation in growth under stress. We did not find any such inter-colonial variation. Rather, we found a strong correlation in growth rates of these lines, even when excluding the known petites ($\tau = 0.31$, $P < 10^{-3}$, see Fig. S1). Furthermore, compensatory mutations are simply spontaneous mutations that should arise at the baseline frequency and effect. The fitness effect of the spontaneous mutations accumulated through 100 bottlenecks were small and both deleterious and beneficial (*Sharp et al., 2018*). By contrast, the changes in fitness that arose during the transformations were of much larger effects and rates. In particular, in our analysis of haploids from DS2, which were transformed with plasmids exclusively to control for potential effects, we observed that 58 out of the 68 transformed haploids transformed with OLP004 turned petite (an astonishing 85%). We speculate that an interaction between the *MATα* locus and heat shock, as its transcription is known to be influenced by temperature (*Sledziewski et al., 1988*; *Manney, Jackson & Meade, 1983*), could explain this pattern.

This study is a cautionary tale. Genetic transformations are commonplace in experimental studies of microbes. We believe that our study highlights the importance of designing experiments such that treatment groups mimic each other as closely as possible, to control for unforeseen effects of laboratory manipulation, and with replication of the transformations themselves, not just replicating fitness assays. Indeed, it initially seemed counterintuitive to us to isolate and use the haploids after plasmid transformation (rather than before). Our initial analysis of DS1, in which haploids were isolated prior to plasmid transformation, showed large differences in the growth rate of haploid and diploid lines. As our results of DS2 show, in which haploids were isolated post-transformation, this effect was driven largely by off-target effects of plasmid transformation. By attempting to hold "all else equal", the method we used introduced variance greater than what we set out to measure. We designed our study with the explicit purpose of measuring small effects, which made the fitness effect of the transformation easy to spot. Even so, it required a close to zero growth rate of a particular line for us to consider the possibility of induced defects. Even within the subset of lines that did not have respiratory defects, we did not find a signal of the mutations we set out to measure. Even with competition experiments assayed with flowcytometric methods *Gallet et al. (2012)* found unexpected variability in their mutational fitness estimates, which they attribute to cryptic variation.

Measuring small mutational effects across ploidy and genotype levels presents a major challenge. A previous attempt to measure the dominance effects of mutations found that heterozygous mutants became homozygous during the fitness assay, literally changing as they were being measured (*Gerstein, Kuzmin & Otto, 2014*). Similarly, in our experiment, yeast lines changed while we transformed them into the desired genotype and ploidy.

## Deletion of *RDH54* protects against petite-formation

We found an unexpected pattern where the *rdh54Δ* deletion mutants were less likely than the wildtype to become respiratory deficient following transformations. Petite yeast are generally more heat tolerant than wildtype yeast (*Van Uden, 1985*). *rdh54Δ* deletion mutants at stationary phase have also been shown to be tolerant to heat shock (*Jarolim et al., 2013*). Even though our strains were subjected to heat shock for transformation in the exponential phase, there may be interactions between temperature, the Rdh54 protein, and mitochondrial function that accounts for the pattern.

A plethora of genes are involved in mitochondrial genome stability (see *Wide-scale screening for rho0 production* in *Contamine & Picard (2000)*). Although there is no evidence as far as we know that the Rdh54 protein is involved in mitochondrial repair or recombination (*Chen, 2013*), the suggestion has been made that petite yeast could arise from homologous recombination between imperfect duplicates (*Faye et al., 1973*; *Gaillard, Strauss & Bernardi, 1980*; *Slonimski & Lazowska, 1977*; *Weiller et al., 1991*); see also the "Recombination/repair track" in *Contamine & Picard (2000)*), and Rdh54 is a vital protein for homologous recombination in the nucleus. Thus another explanation is that the deletion of *RDH54* could reduce mitochondrial recombination rates.

## CONCLUSION

Knowledge of the selection strength of spontaneous mutations in haploid, heterozygous diploids, and homozygous diploids is relevant for the evolution of all sexually reproducing organisms with alternation of ploidy levels. To measure the selective coefficient of small effect mutations we need to generate a large number of mutated genomes in all three genotypic states. Our study shows that the common methods used for transforming yeast into the desired type (ploidy or genotype) led to large changes in fitness that cloud the signal of the mutations of interest and thus make comparisons across ploidy and genotype challenging.

## ACKNOWLEDGEMENTS

The authors would like to thank Dr Nathaniel Sharp for his invaluable contribution to the project design and Matthew Stasiuk, Christina Hsu, and Kismet Somal for assistance in the lab.

### Funding

This project was supported by a Fellowship from the University of British Columbia to Linnea Sandell and a Natural Sciences and Engineering Research Council of Canada grant to Sarah P Otto (RGPIN- 2022-03726). The funders had no role in study design, data collection and analysis, decision to publish, or preparation of the manuscript.

### Grant Disclosures

The following grant information was disclosed by the authors:
University of British Columbia to Linnea Sandell and a Natural Sciences and Engineering Research Council of Canada: RGPIN- 2022-03726.

### Competing Interests

The authors declare that they have no competing interests.

### Author Contributions

- Linnea Sandell conceived and designed the experiments, performed the experiments, analyzed the data, prepared figures and/or tables, authored or reviewed drafts of the article, and approved the final draft.
- Stephan G. König conceived and designed the experiments, performed the experiments, authored or reviewed drafts of the article, and approved the final draft.
- Sarah P Otto conceived and designed the experiments, authored or reviewed drafts of the article, and approved the final draft.

### Data Availability

  The data is available at GitHub and Zenodo:
    - https://github.com/LinneaSandell/schrodingers-yeast.git

- LinneaSandell. (2023). LinneaSandell/schrodingers-yeast: Version at publication (v1.0.0). Zenodo. https://doi.org/10.5281/zenodo.10100606

## Supplemental Information

Supplemental information for this article can be found online at http://dx.doi.org/10.7717/peerj.16547#supplemental-information.

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
