# Peer review of "Schrödinger’s yeast: the challenge of using transformation to compare fitness among Saccharomyces cerevisiae that differ in ploidy or zygosity"

_PeerJ, doi:10.7717/peerj.16547_

## Round 0.1 · original submission · Minor Revisions

Two experts in this field assessed your manuscript and found the content relevant. Some comments need to be addressed before moving this manuscript forward. Among them, the inclusion of more details in the materials and methods section and the inclusion of figures and raw data are requested.

Reviewer 1 ·

Basic reporting

1) abstract has to be rewritten based on the research results, because I could not find out how genetic backgrounds affect spontaneously occurring mutations.
2) could authors please provide more information on delition procedures, e.g. primer sequences?
3) this is a really interesting study, but except for the chemical picture, there is no picture related to the transformation and validation processes.
4) in M&M there is serious lack of cite proper references.

Experimental design

The methods used in this study must be described with sufficient detail and information.

Validity of the findings

the authors are advised to add the "conclusions" part and highlight how the results of this research can be useful for future research.

the authors are recommended to improve the presentation of their results by appropriate figures, graphs or tables

Additional comments

the authors are advised to improve the discussion section by comparing their results with previously published work.

·

Basic reporting

A really interesting manuscript that elaborated on a novel study to compare fitness differences among strains that differ in ploidy and/or zygosity in haploid yeast. Keeping ploidy-specific effects of mutations in mind, the researchers conducted a thorough study to measure mutational effects on fitness in the haploid and homozygous diploid form of yeast. They also give an important cautionary message to other molecular biologists working in this field about using multiple transformants in fitness assays. However, there are a few suggestions:
1) The main findings you got for your objectives are not clearly mentioned in the abstract, instead of mentioning the methods used it is better to elaborate on the results.
2) The writing style followed for the introduction is not as per the journal’s prescribed format
3) Kindly provide a few original images from your experiment that highlight the differences between the control and transformed colonies.
4) If possible, provide the raw data obtained in an Excel file
5) The authors should provide a conclusion

Experimental design

No comments

Validity of the findings

No comments

Additional comments

no comments

---

## Round 0.2 · accepted · Accept

The authors have addressed all the Reviewers' comments. As a consequence, this revised version of the manuscript is suitable for publication.